# Multi-Walled Carbon Nanotube-Induced Gene Expression Biomarkers for Medical and Occupational Surveillance

**DOI:** 10.3390/ijms20112635

**Published:** 2019-05-29

**Authors:** Brandi N. Snyder-Talkington, Chunlin Dong, Salvi Singh, Rebecca Raese, Yong Qian, Dale W. Porter, Michael G. Wolfarth, Nancy L. Guo

**Affiliations:** 1West Virginia University Cancer Institute, West Virginia University, Morgantown, WV 26506, USA; brandi@branditalkington.com (B.N.S.-T.); lindadong2004@yahoo.com (C.D.); ss0083@mix.wvu.edu (S.S.); rebecca.raese@gmail.com (R.R.); 2National Institute for Occupational and Environmental Safety and Health, 1095 Willowdale Rd., Morgantown, WV 26505, USA; yaq2@cdc.gov (Y.Q.); dhp7@cdc.gov (D.W.P.); mgz1@cdc.gov (M.G.W.); 3Department of Occupational and Environmental Health Sciences, School of Public Health, West Virginia University, Morgantown, WV 26506, USA

**Keywords:** multi-walled carbon nanotubes, MWCNT, lung fibrosis, biomarkers

## Abstract

As the demand for multi-walled carbon nanotube (MWCNT) incorporation into industrial and biomedical applications increases, so does the potential for unintentional pulmonary MWCNT exposure, particularly among workers during manufacturing. Pulmonary exposure to MWCNTs raises the potential for development of lung inflammation, fibrosis, and cancer among those exposed; however, there are currently no effective biomarkers for detecting lung fibrosis or predicting the risk of lung cancer resulting from MWCNT exposure. To uncover potential mRNAs and miRNAs that could be used as markers of exposure, this study compared in vivo mRNA and miRNA expression in lung tissue and blood of mice exposed to MWCNTs with in vitro mRNA and miRNA expression from a co-culture model of human lung epithelial and microvascular cells, a system previously shown to have a higher overall genome-scale correlation with mRNA expression in mouse lungs than either cell type grown separately. Concordant mRNAs and miRNAs identified by this study could be used to drive future studies confirming human biomarkers of MWCNT exposure. These potential biomarkers could be used to assess overall worker health and predict the occurrence of MWCNT-induced diseases.

## 1. Introduction

Carbon nanotubes (CNTs) are currently used in diverse industrial and biomedical applications [1,2,3]. With demand driven mainly by increasing applications in manufacturing and decreasing costs of production, the estimated global market value of CNTs is USD 9.85 billion by 2023 [1]. CNT production leads to potential particle exposure in occupational, consumer, and environmental settings, with the highest incidence of unintentional CNT exposure occurring in occupational settings [3]. Particle and dust generation during CNT manufacturing may lead to inhalation exposure to CNTs when these particles become airborne and enter the breathing zone [3,4].

Inhalation of CNTs, particularly multi-walled CNTs (MWCNTs), concentric tubes of rolled graphene, poses a health risk due to their long, rigid, needle-like structure [3]. In humans, workers exposed to MWCNTs during manufacturing were found to have increased reactive oxygen species in exhaled breath condensate as well as elevated profibrotic inflammatory mediators in sputum and serum samples compared with non-exposed workers [4]. Numerous animal studies monitoring the effects of MWCNTs delivered by either intratracheal instillation or inhalation have found increased levels of DNA damage and neutrophil influx following MWCNT exposure [5]. In vitro, exposure of cells found within the pulmonary space, such as alveolar epithelial cells, fibroblasts, and macrophages, to MWCNTs also produces results, such as increases in inflammatory markers, supportive of the toxicity of MWCNT exposure [6,7].

MWCNTs delivered to mice by pharyngeal aspiration deposited in the alveolar region of the lungs and translocated to the subpleural space starting one day after exposure [8]. While this may be a normal clearance mechanism for small particles, longer fibers may not be able to properly migrate through the subpleural space [3,9]. Approximately 65% of the original MWCNT burden remained in the lungs up to 336 days post-exposure; this retention may lead to inflammation and pulmonary fibrosis [3]. While fibrosis is a normal regenerative process following lung injury, insoluble materials that impair the clearance process and lead to a prolonged inflammatory response may result in pulmonary fibrosis, limiting the diffusion capacity for oxygen and resulting in loss of pulmonary function [10]. Additionally, MWCNTs have been categorized by the International Agency for Research on Cancer as group 2B, possibly carcinogenic to humans [11].

While our group has previously found that MWCNT exposure is related to a subset of lung cancer biomarkers in mice following short-term exposure, there are currently no effective biomarkers derived from long-term exposures that are able to detect human lung fibrosis or predict the risk of subsequent lung cancer [12]. Previous studies have shown that animal model-based gene expression profiling can successfully predict human target organ toxicities for numerous human diseases, including cancer [13,14,15,16]. Using in vivo mRNA and miRNA data derived from a study of mice exposed to MWCNT and sacrificed at 1, 6, and 12 months post-exposure [17] and in vitro mRNA and miRNA data derived from a coculture model of human small airway epithelial cells (SAEC) and human microvascular endothelial cells (HMVEC) exposed to MWCNT for 6 and 24 h [18], this study identified mRNAs with concordant MWCNT-induced expression perturbation in in vivo and in vitro studies, identifying them as potential biomarkers for medical and occupational surveillance in humans. The concordant biomarkers identified from this study could be readily tested in human blood samples for monitoring MWCNT-induced disease risk in future studies.

## 2. Results

### 2.1. mRNAs Concordant Between Mouse Tissue and Human Cells

A total of 55 unique mRNAs were identified in mouse lung tissue and SAEC and HMVEC that were concordant regarding changes in gene expression. Table 1 lists the name, expression change direction, and function of the 44 concordant mRNAs between mouse lung tissues at 1, 6, and 12 months post-exposure and SAEC at 6 and 24 h time points. Of these 44 concordant mRNAs, four were upregulated and 40 were downregulated. The fold change values and disease implications for each gene are summarized in Appendix A.

Table 2 lists the name, expression direction, and function of the 24 concordant mRNAs between mouse lung tissue at 1 month, 6 months, and 12 months and HMVEC at any time point. Of these 24 concordant mRNAs, four were upregulated and 20 were downregulated. The fold change values and disease implications for each gene are summarized in Appendix A.

### 2.2. mRNAs Concordant Between Mouse Blood and Human Cells

A total of 15 unique mRNAs were identified in mouse blood and SAEC and HMVEC that were concordant regarding changes in gene expression. Table 3 lists the name, expression direction, and function of the eight concordant mRNAs between mouse blood at 1 month, 6 months, and 12 months and SAEC at any time point. Of these eight concordant mRNAs, one was upregulated and seven were downregulated. The fold change values and disease implications for each gene are summarized in Appendix A.

Table 4 lists the name, expression direction, and function of the eight concordant mRNAs between mouse blood at 1 month and 6 months and HMVEC at any time point. There were no concordant mRNAs between HMVEC and mouse blood at 12 months. Of these eight concordant mRNAs, five were upregulated and three were downregulated. The fold change values and disease implications for each gene are summarized in Appendix A.

### 2.3. mRNAs Concordant between Mouse Lung Tissue, Mouse Blood, and Human Cells

Two mRNAs were found to have concordant expression between mouse lung tissue, mouse blood, and in vitro expression: SLC7A1 and SLC22A5. SLC7A1, solute carrier family 7 member 1, encodes a protein integral to the plasma membrane that is responsible for transport of amino acids across the plasma membrane and is involved in nitric oxide signaling. SLC7A1 was downregulated following MWCNT exposure. SLC22A5, solute carrier family 22 member 5, encodes a protein integral to the plasma membrane that is responsible for organic cation and carnitine transport. SLC22A5 was downregulated following MWCNT exposure.

### 2.4. miRNAs Concordant Between Mouse Tissue and Human Cells

A total of four unique miRNAs were identified in mouse lung tissue and SAEC and HMVEC that were concordant regarding changes in gene expression. Table 5 lists the name of these four concordant miRNAs between mouse lung tissues at 1 month, 6 months, and 12 months and SAEC and HMVEC at any time point. Of these four concordant miRNAs, all were upregulated.

### 2.5. miRNAs Concordant between Mouse Blood and Human Cells

A total of two unique miRNAs were identified in mouse blood and SAEC and HMVEC that were concordant regarding changes in gene expression. Table 6 lists the name of the two concordant miRNAs between mouse blood at 1 month and SAEC or HMVEC. There were no concordant miRNAs between mouse blood at 6 and 12 months postexposure and in vitro expression. All miRNAs were upregulated.

### 2.6. IPA Analysis of Tissue and Cell Concordant mRNAs and miRNAs

All mRNAs and miRNAs concordant between mouse lung tissue and SAEC or HMVEC at any time point were input into Ingenuity Pathway Analysis^®^ (IPA; QIAGEN Inc., https://www.qiagenbioinformatics.com/products/ingenuitypathway-analysis) for analysis of potential cell signaling involvement. The top five molecular and cellular functions associated with the concordant mRNAs are summarized in Figure 1A, while the top five associated diseases and disorders are summarized in Figure 1B. Additionally, these mRNAs were also analyzed for their involvement in diseases and functions previously reported to be related to MWNCT exposure [19]. Several mRNAs listed in Table 1 and Table 2 were found to be associated with fibrosis, inflammatory response, adhesion of neutrophils, and hypertension (Figure 1C); mRNAs that were not currently found to be involved in these processes were removed. To determine potential signaling regulation between tissue and cell concordant mRNAs and miRNAs, all mRNAs and miRNAs concordant between mouse lung tissue and SAEC or HMVEC at any time point were input into IPA and analyzed for potential connections, which are shown in Figure 1D.

### 2.7. IPA Analysis of Blood and Cell Concordant mRNAs

All mRNAs concordant between mouse blood and SAEC or HMVEC at any time point were input into IPA for analysis of potential cell signaling involvement. The top five molecular and cellular functions associated with these concordant mRNAs are shown in Figure 2A, while the top five associated diseases and disorders are summarized in Figure 2B. The relationship between the mRNAs listed in Table 3 and Table 4 with known MWCNT-related diseases and functions are summarized in Figure 2C; mRNAs that were not currently found to be involved in these processes were removed. No mRNAs were found to be involved in adhesion of neutrophils or hypertension. To determine potential signaling regulation between blood and cell concordant mRNAs and miRNAs, all mRNAs and miRNAs concordant between mouse blood and SAEC or HMVEC at any time point were input into IPA and analyzed for potential connections. These potential regulatory connections are shown in Figure 2D.

## 3. Discussion

This study identified mRNAs and miRNAs with concordant MWCNT-induced expression perturbation between in vivo and in vitro settings. We previously reported that MWCNT-induced mRNA expression in SAEC and HMVEC grown in co-culture has higher overall genome-scale correlation with mRNA expression in mouse lungs following MWCNT inhalation than either cell type grown separately in culture [18]; therefore, the mRNAs and miRNAs identified in this study may serve as potential biomarkers for medical and occupational surveillance in humans in future studies.

Overall, tissue and cell concordant mRNAs as well as the blood and cell concordant mRNAs are involved in many cellular processes and have varied disease implications. Two mRNAs were concordant between all tissue, blood, and cell analyses: SLC7A1 and SLC22A5, both of which were found to be downregulated following MWCNT exposure.

The *SLC7A1* gene encodes the CAT-1 protein, a transmembrane protein involved in the transport of amino acids, particularly arginine; CAT-1 is part of the nitric oxide signaling pathway [20]. CAT-1 plays an important role in the influx of L-arginine, a substrate necessary for proper eNOS function. When *SLC7A1* functionality, and thus arginine transport, is lost, nitric oxide production can become abnormal, potentially resulting in certain vascular disorders, such as hypertension [20]. In a study by Chen et al. (2015), instillation of MWCNTs into rats resulted in an increase of markers indicative of numerous cardiovascular diseases, including hypertension [21]. Chen et al. (2015) noted that this effect was particularly noticeable in spontaneously hypertensive rats, which are considered a good model of human primary hypertension [21]. Several in vivo studies have also shown that pulmonary exposure to MWCNTs induces impairment of arteriolar dilation, vascular dysfunction, and cardiac ischemia/reperfusion injury [22,23,24,25], and a human population study showed an association between workers exposed to MWCNTs and ICAM-1, a cardiovascular biomarker [26]. Recently, a study demonstrated that inhalation of MWCNTs significantly increased both systolic and diastolic blood pressure and decreased heart rate in rats [27]. Coupled together, these results and our outcomes suggest that mRNA expression levels of *SLC7A1* may serve as both a tissue and blood biomarker of hypertension in workers exposed to MWCNTs and may be particularly useful in workers with preexisting cardiovascular disease.

The *SLC22A5* gene encodes the OCTN2 protein, an integral membrane protein that functions as both an organic cation transporter and a sodium-dependent carnitine transporter. Downregulation of *SLC22A5* can induce primary carnitine deficiency, which is characterized by encephalopathy, cardiomyopathy, cardiomegaly, metabolic derangement, hypoglycemia, and muscle weakness [28,29]. Reduced OCTN2 levels have also been noted in cellular studies of cancer, while OCTN2 knockout mice have systemic immune activation and inflammation [29,30]. Like *SLC7A1*, tissue and blood expression levels of *SLC22A5* may indicate a MWCNT-induced disease state and can potentially be used as a biomarker of disease.

IPA analysis of mRNAs concordant between cell and mouse lung tissue analyses identified their top five related molecular and cellular functions to be cellular assembly and organization; DNA replication, recombination, and repair; carbohydrate metabolism, lipid metabolism, and small molecule biochemistry. Examining these functions in more detail, MWCNT exposure caused perturbation of genes involved in chromosomal aggregation, catabolism of DNA, and phosphatidylinositol 3,5-diphosphate synthesis. Cellular exposure to MWCNT at occupationally relevant doses has been shown to disrupt mitosis and result in polyploidy and other genetic damage [31]. IPA analysis also identified the top five diseases and disorders associated with cell and lung tissue concordant mRNA to be hematological disease, cancer, gastrointestinal disease, hepatic system disease, and organismal injury and abnormalities. The effects of these mRNAs on these diseases and disorders also suggest potential MWCNT-induced chromosome and DNA damage, which is consistent with previously reported genotoxicity and carcinogenicity of MWCNTs [31,32]. A study demonstrated that MWCNTs promote growth and neoplastic progression of initiated lung cells in mice following inhalation exposure [33]. Moreover, inhalation of MWCNTs significantly increased lung carcinomas and combined carcinoma and adenoma formation in mice in a 104-week carcinogenicity study [32].

Of the tissue and cell concordant mRNAs, CAND1 was found to be associated with hypertension, while IL1R1 was found to be associated with adhesion of neutrophils and fibrosis and IL1R1 in conjunction with TGM2 was associated with an overall inflammatory response, all of which are outcomes attributed to MWCNT exposure. When incorporating both cell and tissue concordant mRNAs and miRNAs, miRNA/mRNA interactions were found between miR-183 and SEL1L and SLC30A7; miR-26b and MAN2B2; and miR-204 and NPM1. MiR-183, miR-26b, and miR-204 have all been implicated in the progression of cancer, and their expression and subsequent effects on signaling pathways may serve as miRNA biomarkers of MWCNT exposure and potential disease [34,35,36].

IPA analysis of mRNAs concordant between cell and mouse blood analyses identified their top five related molecular and cellular functions to be lipid metabolism, small molecule biochemistry, vitamin and mineral metabolism, amino acid metabolism, and cell morphology. When looking at these functions in more detail, MWCNT exposure caused perturbation of genes involved in cellular metabolism and cellular morphology, particularly overall cell size and the size of the centrosome. As suggested with the cell and tissue concordant mRNA molecular and cellular functions, perturbations in mRNA expression in blood following MWCNT exposure may suggest DNA damage and overall loss of cellular integrity. IPA analysis also identified the top five diseases and disorders associated with cell and lung tissue concordant mRNA to be developmental disorder, hereditary disorder, metabolic disorder, organismal injury and abnormalities, and respiratory disease. When considered together, aberrant mRNA expression in these categories was associated with cystic fibrosis, suggesting that these mRNAs can be surveyed in blood and potentially act as biomarkers of lung fibrosis.

Of the cell and blood concordant mRNAs, CFTR was found to be associated with fibrosis and an inflammatory response, while potential miRNA/mRNA regulation was found between miR-148a-3p and SCL22A5. As with the regulatory pathways from the cell and tissue concordant analysis, these mRNAs and miRNAs and their potential signaling pathways may serve as blood biomarkers of MWCNT exposure and disease.

There are currently no effective biomarkers to detect human lung fibrosis or predict the risk of subsequent lung cancer resulting from MWCNT exposure. The mRNAs and miRNAs and their potential signaling pathways identified in this study from both mouse lung tissue and blood could possibly be used in future studies to confirm biomarkers of MWCNT exposure in the workplace. These biomarkers could not only be used to assess overall worker health but also predict the occurrence of MWCNT-induced disease.

## 4. Materials and Methods

### 4.1. MWCNT

The MWCNT used in this study has been previously described [17]. Briefly, the MWCNT were obtained from Mitsui & Company (MWNT-7, lot #05072001K28) and manufactured using a floating reactant catalytic chemical vapor deposition method followed by high-temperature thermal treatment in argon at 2500 °C using a continuous furnace [37]. Extensive characterization of the MWCNT was previously published in Reference [38]. In summary, the number of walls ranged between 20 and 50, median length was 3.86 ± 1.9 μm, and count mean width was 49 ± 13.4 nm. Trace metal contamination was 0.78%, with sodium (Na, 0.41%) and iron (Fe, 0.32%) being the two major metal contaminants [38]. Endotoxin levels in the MWCNT were below the levels of detection [38].

### 4.2. In Vivo Studies

The animal study from which the in vivo data for this study was derived has been previously reported in Reference [17]. Briefly, Male C57BL/J6 mice (7 weeks old, average weight 21.19 g ± 0.06 g) were obtained from Jackson Laboratories (Bar Harbor, ME). The study consisted of three components: bronchoalveolar lavage (BAL), pulmonary histopathology, and lung tissue mRNA microarray analysis. Each component consisted of eight mice per exposure group (dispersion media [DM, negative control]; 1, 10, 40, or 80 µg MWCNT; 120 µg crocidolite asbestos [positive control]) to be sacrificed at 1, 6 or 12 months post-exposure, for a total of 144 mice per each arm of BAL, pulmonary histopathology, and lung tissue mRNA microarray analysis and 432 mice for the total study. Rationale for the dose selection of MWCNT has been previously described in Reference [38].

### 4.3. In Vivo mRNA Microarray Processing

Global mRNA expression profiles were generated with Mouse Gene ST 2.1 plates at the University of Michigan Microarray Facility using an Affymetrix Plus kit. cDNA was synthesized from 500 ng total RNA, amplified, fragmented, and biotinylated using a GeneChip WT PLUS Reagent kit (Affymetrix; Santa Clara, CA, USA) according to manufacturer’s instructions. cDNA was then prepared for hybridization with reagents from the Affymetrix GeneTitan Hybridization, Wash, and Stain Kit for WT Array Plates. For hybridization, 2.76 µg labeled cDNA were hybridized to the Affymetrix Mouse Gene ST 2.1 Arrays, which were then washed, stained, and scanned using the GeneTitan Multi-Channel Instrument according to Affymetrix’s User Guide for Expression Array Plates (P/N 702933 Rev. 2, 2013). Data were analyzed using the Limma, Oligo, and Affy Bioconductor packages implemented in the R statistical environment. The robust multi-array average was used to normalize the data and fit log_2_ transformed expression values. The raw microarray data were processed using a robust multi-array average method, and expression values were log_2_ transformed, with a principal component analysis utilized as the final quality control step to visualize mRNA expression values [39]. The analysis was performed with the oligo package of Bioconductor in the R statistical environment. The mRNA expression profiles in mouse blood are available in NCBI Gene Expression Omnibus (GEO) with the accession number GSE126959.

### 4.4. In Vitro Studies

The cellular study from which the in vitro data for this investigation was derived has been previously described in Reference [18]. Briefly, for monoculture exposures, SAEC and HMVEC were plated directly into 100 mm cell culture dishes (Corning, Tewksbury, MA, USA; growth area: 55 cm^2^), allowed to form intact barriers for 72 h, serum starved overnight, and exposed directly to MWCNT at a concentration of 1.2 µg/mL in 10 mL of their respective media for either 6 or 24 h. DM for 24 h was used as a negative control. For co-culture exposures, SAEC were seeded onto apical transwell chambers while HMVEC were planted into the basolateral chambers of a 6-well transwell dish. SAEC and HMVEC were allowed to form intact epithelial and endothelial barriers for 72 h, serum starved overnight, and SAEC were exposed to MWCNT at a concentration of 1.2 µg/mL in 10 mL of SAEC media for either 6 or 24 h. DM for 24 h was used as a negative control. HMVEC in the coculture system were not directly exposed to MWCNT, and MWCNT are not apparent in HMVEC transmission electron microscopy preparations following SAEC exposure in co-culture.

### 4.5. In Vitro Microarray mRNA Profiling

RNA samples were analyzed for expression profiling using Agilent G3 Human Gene Expression 8 × 60 k Arrays (Agilent; Santa Clara, CA, USA). Total RNA quality for both microarray analyses was determined on an Agilent 2100 Bioanalyzer, with all samples having RNA integrity numbers greater than 8. Total RNA (250 ng) was used for labeling using a QuickAmp labeling kit (Agilent). Extracted RNA was labeled with cyanine (Cy)-3-CTP (PerkinElmer; Waltham, MA, USA) and reference RNA with (Cy)-5-CTP. Following purification of labeled cRNAs, 825 ng of Cy3- and Cy5-labeled cRNAs were combined and hybridized for 17 h at 65 °C in an Agilent hybridization oven. Microarrays were then washed and scanned using an Agilent DNA Microarray Scanner. The mRNA expression profiles in SAEC and HMVEC couture and monoculture are available in NCBI GEO with the accession number GSE129640.

### 4.6. miRNA Expression Profiling and Processing

For in vivo animal blood miRNA profiling, RNA samples were sent to Ocean Ridge Biosciences (Palm Beach Gardens, FL, USA) for analysis using custom, multi-species microarrays containing 10,290 total features present in miRBase version 19. For in vitro cell studies, total RNA was analyzed using custom, multi-species microarrays containing 8817 human miRNA total features present in miRBase version 17 by Ocean Ridge Biosciences (Palm Beach Gardens, FL, USA). The sensitivity of the microarray is such that it could detect as low as 20 moles of synthetic miRNA being hybridized along with each sample. The microarrays were produced by Microarrays Inc. (Huntsville, Alabama) and consisted of epoxide glass substrates that had been spotted in triplicate with each probe. Quality control of the total RNA samples was assessed using UV spectrophotometry and agarose gel electrophoresis. The samples were DNAse digested, and low-molecular weight (LMW) RNA was isolated by ultrafiltration through YM-100 columns (Millipore; St Louis, MO, USA) and subsequently purified using a RNeasy MinElute Clean-Up Kit (Qiagen; Germantown, MD, USA). The LMW RNA samples were 3′-end labeled with Oyster-550 fluorescent dye using a Flash Taq RNA Labeling Kit (Genisphere; Hatfield, PA, USA). Labeled LMW RNA samples were hybridized to the miRNA microarrays according to conditions recommended in the Flash Taq RNA Labeling Kit manual. The microarrays were scanned on an Axon Genepix 4000B scanner, and data were extracted from images using GenePix V4.1 software.

Spot intensities were obtained for all the features on each microarray by subtracting the median local background from the median local foreground for each spot. Detection Thresholds (T) for each array were determined by calculating the 10% trim mean intensity of the negative control spots and adding 5× the standard deviation of the background (non-spot area). The spot intensities and the T were transformed by taking the log_2_ base 2 of each value. The normalization factor (N) for each microarray was determined by obtaining the 20% trim mean of the spot intensities for the mouse probes above threshold in all samples. The log_2_-transformed spot intensities for all the features were normalized by subtracting N from each spot intensity and scaled by adding the grand mean of N across all microarrays. The mean probe intensities for each of the 1936 mouse probes on each of the 160 arrays (one sample per array) were then determined by averaging the triplicate spot intensities. Spots flagged as poor quality during data extraction were omitted prior to averaging. The 704 mouse non-control, log_2_-transformed, normalized, and averaged probe processing intensities were filtered to obtain a list of 484 mouse miRNA probes showing probe intensity above T in all samples from at least one treatment group.

After quality control and preprocessing, 385 detectable mouse miRNA probes from the in vivo studies, and 555 human miRNA probes in HMVEC and 543 human miRNA probes in SAEC from the in vitro studies, were detected for further analysis. The miRNA expression profiles are available in the NCBI GEO database with accession numbers GSE130109 (mouse blood miRNA), GSE130790 (mouse lung miRNA), and GSE13123 (cellular miRNA).

### 4.7. In Vivo mRNA/miRNA Microarray Data Analysis

The mouse mRNA microarray consisted of 41,345 probes. Variance was calculated for each probe across all samples to remove any probe with variance >0.2. All probes resulted in variance <0.2; therefore, no probes were removed due to variance. Unknowns were then removed from the data, leaving 25,616 probes. Data analysis was performed with 26,191 probes. ANOVA was used to compare each treatment group at each post-exposure time point to its respective DM control to determine significantly up- and down-regulated mRNAs with a *p*-value <0.05, false discovery rate (FDR) of 10%, and fold change (FC) ≥1.5. Similar analysis was applied to miRNA microarray data analysis.

### 4.8. In Vitro mRNA/miRNA Microarray Data Analysis

In vitro microarray data were exported using Feature Extraction v10 as tab-delimited text files after background subtraction, log_2_ transformation, and lowess normalization and reported as log or relative expression of sample compared to a universal reference. Data were read from each file into R using a custom script [40]. For each array, values for control spots, spots that were saturated on either channel, spots that were reported by Feature Extraction as non-uniform outliers on either channel, and spots which were not well above background on at least one channel were considered unreliable and/or uninformative and replaced by “NA.” Values were collated into a single table, and probes for which fewer than ten present values were available were removed. For probes spotted multiple times on the array, values were averaged across replicate probes.

Missing data were imputed using the K-means nearest neighbor algorithm as implemented by the impute.knn function in the impute R package from Bioconductor (Bioconductor; Seattle, WA, USA). For each dose and time point, a set of differentially expressed genes was identified by performing a two-class unpaired Significance Analysis of Microarrays (SAM) between the treated samples and the dose zero samples from the corresponding time point, using the Bioconductor package. A threshold delta value was chosen to produce a false discovery rate of 10% using the findDelta function from the same package. The list of probes called as significant were subsequently filtered by restricting to those probes which were at least 1.5-fold up- or down-regulated (fold changes were computed from the data before imputation of missing values).

### 4.9. Identification of Concordant Genes from In Vitro and In Vivo Studies

The mouse and human genomes were matched by gene name using the data mining tool Biomart based on Ensembl Genes version 68 (http://www.ensembl.org/biomart/martview/) [41]. There was a total of 15,473 matched genes (orthologs) between human and mouse. For ortholog genes defined as a “one-to-many” or “many-to-many” relationship between the mouse and human genome, a randomly selected matched gene pair was chosen. Significant genes from in vitro human cells under different treatment conditions and in vivo mouse lung and blood were matched and identified.

## Figures and Tables

**Figure 1 ijms-20-02635-f001:**
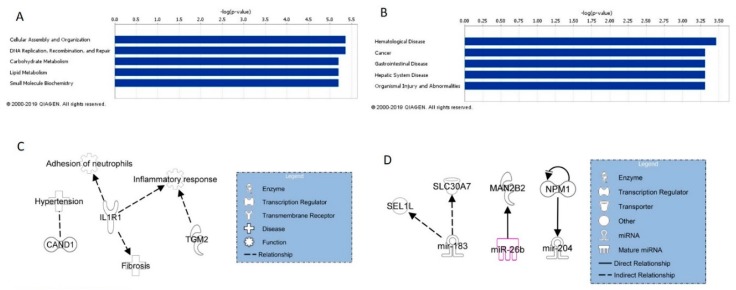
Tissue and cell concordant mRNA and miRNA IPA analysis. (**A**) Top five molecular and cellular functions associated with concordant mRNAs. (**B**) Top five diseases and disorders associated with concordant mRNAs. (**C**) mRNAs associated with known MWCNT exposure outcomes. (**D**) mRNA and miRNA regulation in tissue and cell concordant mRNAs and miRNAs.

**Figure 2 ijms-20-02635-f002:**
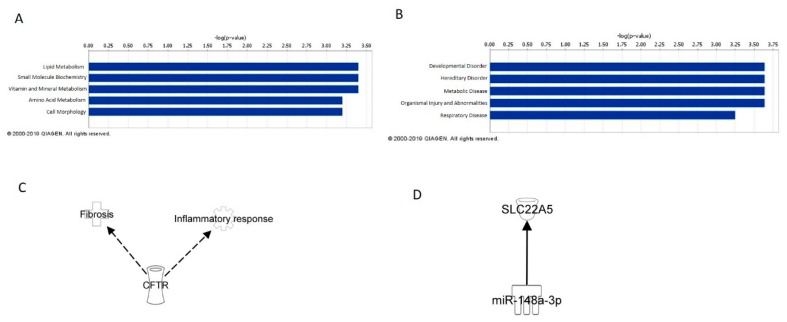
Blood and Cell Concordant mRNA and miRNA IPA Analysis. (**A**) Top five molecular and cellular functions associated with concordant mRNAs. (**B**) Top five diseases and disorders associated with concordant mRNAs. (**C**) mRNAs associated with known MWCNT exposure outcomes. (**D**) mRNA and miRNA regulation in blood and cell concordant mRNAs and miRNAs.

**Table 1 ijms-20-02635-t001:** Concordant mRNAs in mouse tissue and human small airway epithelial cells (SAEC).

mRNA	Tissue and Cell Regulation	Function
**1-month animal exposure**		
CAND1	Downregulated	Ubiquitin ligase regulation
CYCS	Downregulated	Electron transport chain in mitochondria
DENND5B	Downregulated	Calcium ion transmembrane transport
HS3ST3B1	Downregulated	Sulfotransferase
MAML1	Downregulated	Cell fate determination
MAN2B2	Downregulated	Hydrolase
METTL21	Downregulated	Methyltransferase
PIGS	Downregulated	GPI-anchor biosynthesis
PPP1C	Downregulated	Phosphatase
S100A5	Upregulated	Calcium binding
SEL1L	Downregulated	Misfolded protein translocation
SH3BP2	Downregulated	Adaptor protein
SLC7A1	Downregulated	Transmembrane transporter
TGM2	Downregulated	Transglutaminase
XRN2	Downregulated	Exonuclease
**6-month animal exposure**		
KIF14	Downregulated	Microtubule motor protein
MYBPC2	Upregulated	Myosin binding
PSD4	Upregulated	ARF protein signal transduction
TTLL7	Downregulated	Cell differentiation
**12-month animal exposure**		
ADH5	Downregulated	Alcohol dehydrogenase
ANLN	Downregulated	Cell growth and migration
ARF1	Downregulated	Guanine nucleotide binding
CCDC115	Downregulated	Unfolded protein binding
CYCS	Downregulated	Electron transport chain in mitochondria
DDX24	Downregulated	RNA helicase
FAM188A	Downregulated	Apoptosis
IKZF2	Downregulated	Zinc finger protein
IL1R1	Downregulated	Interleukin receptor
INTS4	Downregulated	snRNA processing
MAD2L1	Downregulated	Mitotic spindle assembly checkpoint
PIGU	Downregulated	Cell division control
PIKFYVE	Downregulated	Cytoskeletal functions, membrane trafficking, and receptor signaling
PNRC2	Downregulated	DNA/mRNA binding
PRKRIR	Downregulated	Regulation of cell proliferation
PRIM2	Downregulated	Replication of DNA
RNF19A	Downregulated	Ubiquitin ligase
SDHD	Downregulated	Succinate oxidation
SGOL2	Downregulated	Cell cycle regulation
SLC30A7	Downregulated	Zinc transporter
TMPO	Downregulated	Nuclear organization
TMPRSS6	Upregulated	Serine proteinase
UBE2E1	Downregulated	Ubiquitination
VAC14	Downregulated	Regulates levels of phosphatidylinositol 3,5-bisphosphate
WDR74	Downregulated	RNA regulation

**Table 2 ijms-20-02635-t002:** Concordant mRNAs in mouse tissue and human microvascular endothelial cells (HMVEC).

mRNA	Tissue and Cell Regulation	Function
**1-month animal exposure**		
CAND1	Downregulated	Ubiquitin ligase regulation
HIST1H3F	Upregulated	DNA structure
PPP1CC	Downregulated	Phosphatase
SH2D1A	Downregulated	T and B cell stimulation
SH3BP2	Downregulated	Adaptor protein
SLC22A5	Downregulated	Glucose transporter
TGM2	Downregulated	Transglutaminase
**6-month animal exposure**		
HIST1H2AL	Upregulated	DNA structure
TFEC	Downregulated	Transcription factor
**12-month animal exposure**		
ABCE1	Downregulated	Molecular transport
EIF4B	Downregulated	Helicase
IKZF2	Downregulated	Transcription factor
IL1R1	Downregulated	Interleukin receptor
LXN	Downregulated	Metallocarboxypeptidase inhibition
MID1	Upregulated	Adaptor protein
NPM1	Downregulated	ARF/p53 regulation
NUDT8	Upregulated	Hydrolase
PIKFYVE	Downregulated	Cytoskeletal functions, membrane trafficking, and receptor signaling
PPP1CC	Downregulated	Phosphatase
SDHD	Downregulated	Succinate oxidation
SLC30A7	Downregulated	Zinc transporter
SMARCAD1	Downregulated	Helicase
UBE2E1	Downregulated	Ubiquitination
ZC3H13	Downregulated	mRNA methylation

**Table 3 ijms-20-02635-t003:** Concordant mRNAs in mouse blood and SAEC.

mRNA	Blood and Cell Regulation	Function
**1-month animal exposure**		
CFTR	Downregulated	Multidrug resistance
HSD17B2	Upregulated	Bone development
PER1	Downregulated	Circadian rhythm
RIMKLB	Downregulated	Amino acid synthesis
SH3RF3	Downregulated	Metal ion binding
SLC7A1	Downregulated	Amino acid transport
**6-month animal exposure**		
HMGB2	Downregulated	DNA bending
**12-month animal exposure**		
SYTL3	Downregulated	Vesicle trafficking

**Table 4 ijms-20-02635-t004:** Concordant mRNAs in mouse blood and HMVEC.

mRNA	Blood and Cell Regulation	Function
**1-month animal exposure**		
PER1	Downregulated	Circadian rhythm
SLC22A5	Downregulated	Glucose transporter
TUBA1B	Upregulated	Cell division
**6-month animal exposure**		
HIST1H3H	Upregulated	Chromosomal structure
KPRP	Upregulated	Keratinocyte differentiation
KRT79	Upregulated	Epithelial cell integrity
MID1	Downregulated	Adaptor protein
SHCBP1	Upregulated	Cell proliferation

**Table 5 ijms-20-02635-t005:** Concordant miRNAs in mouse tissue and SAEC or HMVEC.

1-Month Animal Exposure	6-Month Animal Exposure	12-Month Animal Exposure
**SAEC**		
miR-183	miR-204	miR-335
**HMVEC**		
None	miR-204	miR-335; miR-26b

**Table 6 ijms-20-02635-t006:** Concordant miRNAs in mouse blood and SAEC.

1-Month Animal Exposure	6-Month Animal Exposure	12-Month Animal Exposure
**SAEC**		
miR-148b	None	None
**HMVEC**		
miR-29c	None	None

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
