# Peer review of "Multi-Walled Carbon Nanotube-Induced Gene Expression Biomarkers for Medical and Occupational Surveillance"

_ijms, 2019, doi:10.3390/ijms20112635_

Round 1
Reviewer 1 Report
Good Work.
In the manuscript ‘Multi-walled Carbon Nanotube-Induced Gene Expression Biomarkers for Medical and Occupational Surveillance’, authors have described analyzing the biomarkers to assess health of the workers working with multi wall carbon nano tubes and predict the occurrence of diseases. This is an important research as use and applications of MWCNT are increasing, and yet their exposure related problems remain a challenge.
After reading the manuscript thoroughly, my comments are:
· The paper is well written and well explained
· This study is important from application point of view
· Work has been supported by enough relevant literature
· There are hardly any questions in relevance to this study that have remained unanswered
· There are no grammatical errors in the manuscript.
This paper is acceptable without any changes.
Author Response
Reviewer 1
In the manuscript ‘Multi-walled Carbon Nanotube-Induced Gene Expression Biomarkers for Medical and Occupational Surveillance’, authors have described analyzing the biomarkers to assess health of the workers working with multi wall carbon nano tubes and predict the occurrence of diseases. This is an important research as use and applications of MWCNT are increasing, and yet their exposure related problems remain a challenge.
After reading the manuscript thoroughly, my comments are:
·The paper is well written and well explained
·This study is important from application point of view
·Work has been supported by enough relevant literature
·There are hardly any questions in relevance to this study that have remained unanswered
·There are no grammatical errors in the manuscript.
This paper is acceptable without any changes.
Authors: We thank the reviewer for thorough review of our manuscript.
Reviewer 2 Report
The article entitled” Multi-walled Carbon Nanotube-Induced Gene Expression Biomarkers for Medical and Occupational Surveillance” is a potentially interesting work, but there are a number of points which should be considered:
1. The introduction must be reworded in order to facilitate the readers. I suggest to add the following recent and interesting references that focalized in details the toxicity theme:
- Pulmonary toxicity of two different multi-walled carbon nanotubes in rat: Comparison between intratracheal instillation and inhalation exposure, by Gatè et al. 2019 (https://www.sciencedirect.com/science/article/pii/S0041008X19301644)
- Profibrotic Activity of Multiwalled Carbon Nanotubes Upon Prolonged Exposures in Different Human Lung Cell Types by Chortarea et al. 2019
https://www.liebertpub.com/doi/full/10.1089/aivt.2017.0033
-Toxicity Assessment in the Nanoparticle Era BY De Matteis V 2018 https://link.springer.com/chapter/10.1007%2F978-3-319-72041-8_1
-Multi-Walled Carbon Nanotube-Induced Gene Expression in the Mouse Lung: Association with Lung Pathology by Pacurari et al 2011
https://www.sciencedirect.com/science/article/pii/S0041008X11001918?via%3Dihub
2. Characterization of MWCNTs: I would suggest to add a TEM or SEM image of nanomaterials.
3. all the images are very poor quality and the reading of the histograms is difficult. I suggest to the authors to replace images with others good quality and resolution images.
4.Dose selection for in vitro study: The rationale for dose selection is absent.
5. The authors need to check the English
Author Response
Reviewer 2
1. The introduction must be reworded in order to facilitate the readers. I suggest to add the following recent and interesting references that focalized in details the toxicity theme:
- Pulmonary toxicity of two different multi-walled carbon nanotubes in rat: Comparison between intratracheal instillation and inhalation exposure, by Gatè et al. 2019 (https://www.sciencedirect.com/science/article/pii/S0041008X19301644)
- Profibrotic Activity of Multiwalled Carbon Nanotubes Upon Prolonged Exposures in Different Human Lung Cell Types by Chortarea et al. 2019
https://www.liebertpub.com/doi/full/10.1089/aivt.2017.0033
-Toxicity Assessment in the Nanoparticle Era BY De Matteis V 2018 https://link.springer.com/chapter/10.1007%2F978-3-319-72041-8_1
-Multi-Walled Carbon Nanotube-Induced Gene Expression in the Mouse Lung: Association with Lung Pathology by Pacurari et al 2011
https://www.sciencedirect.com/science/article/pii/S0041008X11001918?via%3Dihub
Authors: We have expanded the introduction and added these references.
Reviewer 2
2. Characterization of MWCNTs: I would suggest to add a TEM or SEM image of nanomaterials.
Authors: TEM images of the MWCNTs used in this study have been previously published: Porter et al. Mouse pulmonary dose- and time course-responses induced by exposure to multi-walled carbon nanotubes. Toxicology. 2010;269(2-3):136-147.
FESEM images of the MWCNTs used in this study in mouse lungs following pulmonary aspiration exposure have been previously published: Porter et al. Mouse pulmonary dose- and time course-responses induced by exposure to multi-walled carbon nanotubes. Toxicology. 2010;269(2-3):136-147.
TEM images of the MWCNTs used in this study in the coculture system have been previously published: Snyder-Talkington, et al. Multi-walled carbon nanotubes induce human microvascular endothelial cellular effects in an alveolar-capillary co-culture with small airway epithelial cells. Part Fibre Toxicol. 2013;10:35.
Reviewer 2
3. all the images are very poor quality and the reading of the histograms is difficult. I suggest to the authors to replace images with others good quality and resolution images.
Authors: High quality images have been provided.
Reviewer 2
4.Dose selection for in vitro study: The rationale for dose selection is absent.
Authors: Rationale for the dose selection in this study has been published in part: Porter et al. Mouse pulmonary dose- and time course-responses induced by exposure to multi-walled carbon nanotubes. Toxicology. 2010;269(2-3):136-147.
Briefly,
In order to evaluate the relevance of the findings of this in vivo mouse study to human MWCNT exposures, we need to determine if the doses tested in mice are relevant human occupational exposures. Assuming amouse alveolar epithelium surface area of 0.05 m2 (Stone et al., 1992), the 10 g MWCNT dose would result in 200 g MWCNT/m2 alveolar epithelium, whereas the 80 g MWCNT would result in 1600 g MWCNT/m2 alveolar epithelium. A recent study reported peak MWCNT-containing airborne dust levels of approximately 400 g/m3 in a research laboratory (Han et al., 2008). Assuming a peak MWCNT aerosol of 400 g/m3, MWCNT mass median aerodynamic diameter (MMAD) = 1.5 m (Porter et al., 2009), minute ventilation of 20 l/min (Galer et al., 1992) and deposition fraction of 30% (Phalen, 1984), and human alveolar epithelium surface area of 102 m2 (Stone et al., 1992), approximate human exposure per month would be 226 g MWCNT/m2 alveolar epithelium. Thus, 10 g MWCNT exposure in mouse approximates human deposition for a person performing light work for one month in a work environment with MWCNT aerosol of 400 g/m3. Even if the average daily MWCNT aerosol is determined to be much lower once more workplace exposure data is collected, e.g., 4–40 g/m3, the 10 g MWCNT exposure in mouse would approximate human deposition for a person performing light work for approximately 9 months to 7.5 years. Thus, these estimates suggest that the MWCNT doses tested in mice in this study approximate reasonable human occupational exposures to MWCNT.
In this current study, we also added an exposure of MWCNT at 1 μg to study the effects of a lower dose of MWCNT. We have added this reference to the Materials and Methods.
Reviewer 2
5. The authors need to check the English
Authors: We have conducted a thorough review of the manuscript.
Reviewer 3 Report
This article investigated mRNA with concordant MWCNT-induced expression perturbation in in vivo and in vitro trials, thus trying to identify potential biomarkers for medical and occupational surveillance in humans. The discovery of MWCNT-specific biomarkers for human exposure to MWCNTs is important. However, there are some critical points that require attention, as follows:
1. The reviewer could not validate all array data that the authors referred to in their paper; nevertheless, the reviewer could confirm some mRNA array data from reference 13. The data sheets in the supplemental file 1 showed altered mRNA expression when tissues were exposed to four MWCNT concentrations. However, most mRNA samples did not show a concentration dependency. Despite this, the authors used all of the genes in their study. Moreover, the mRNA samples in Table 1 and 2 do not show a time dependency. Basically, then, it appears that the main purpose of the mRNA expression assays by microarray was screening. The authors should show the quantified results for Tables 1-6 by RT-PCR or another quantification method.
2. This paper’s title is “Multi-walled (incidentally, “walled” should probably be capitalized for consistency) Carbon Nanotube-Induced Gene Expression Biomarkers for Medical and Occupational Surveillance”. However, the authors do not specify any mRNA or miRNA names as biomarkers. The authors state that they do not know whether the listed mRNA and miRNA results are specific for MWCNT exposure. They should at least validate the data using the results of asbestos described as a positive control.
3. The MWNT-7 of lot number 05072001K28 was not produced by Hodogaya Chemical Company. The authors should confirm the accuracy of this and other information.
Author Response
Reviewer 3
This article investigated mRNA with concordant MWCNT-induced expression perturbation in in vivoand in vitro trials, thus trying to identify potential biomarkers for medical and occupational surveillance in humans. The discovery of MWCNT-specific biomarkers for human exposure to MWCNTs is important. However, there are some critical points that require attention, as follows:
1. The reviewer could not validate all array data that the authors referred to in their paper; nevertheless, the reviewer could confirm some mRNA array data from reference 13. The data sheets in the supplemental file 1 showed altered mRNA expression when tissues were exposed to four MWCNT concentrations. However, most mRNA samples did not show a concentration dependency. Despite this, the authors used all of the genes in their study. Moreover, the mRNA samples in Table 1 and 2 do not show a time dependency. Basically, then, it appears that the main purpose of the mRNA expression assays by microarray was screening. The authors should show the quantified results for Tables 1-6 by RT-PCR or another quantification method.
Authors: The reviewer is correct that the initial purpose of this microarray analysis is screening for potential biomarkers for use in medical and occupational surveillance. Global Profiling for identification of concordant genes from animal lung tissues/blood and human cell lines provides a two-layered approach: in vivo/in vitro screening and validation. While the qRT-PCR of the identified concordant genes in blood/biopsy samples from MWCNT-exposed workers will be a focus of our future work, it is currently outside the scope of this analysis.
Additionally, we provide the reviewer with the following GEO accession numbers and have added these to the paper:
GSE130109: mouse blood miRNA
GSE130790: mouse lung miRNA
GSE131123: cell miRNA
GSE129640: two cell mRNA
GSE126959: mouse blood mRNA
Reviewer 3
2. This paper’s title is “Multi-walled (incidentally, “walled” should probably be capitalized for consistency) Carbon Nanotube-Induced Gene Expression Biomarkers for Medical and Occupational Surveillance”. However, the authors do not specify any mRNA or miRNA names as biomarkers. The authors state that they do not know whether the listed mRNA and miRNA results are specific for MWCNT exposure. They should at least validate the data using the results of asbestos described as a positive control.
Authors: It was not our intent to suggest that the listed mRNA and miRNA results are not specific for MWCNT exposure and have reworded these portions of the manuscript.
Reviewer 3
3. The MWNT-7 of lot number 05072001K28 was not produced by Hodogaya Chemical Company. The authors should confirm the accuracy of this and other information.
Authors: We have corrected this.
Round 2
Reviewer 3 Report
I agree to the publication of this manuscript.